# Aquatic Training after Joint Immobilization in Rats Promotes Adaptations in Myotendinous Junctions

**DOI:** 10.3390/ijms22136983

**Published:** 2021-06-29

**Authors:** Lara Caetano Rocha, Gabriela Klein Barbosa, Jurandyr Pimentel Neto, Carolina dos Santos Jacob, Andreas B. Knudsen, Ii-Sei Watanabe, Adriano Polican Ciena

**Affiliations:** 1Laboratory of Morphology and Physical Activity (LAMAF), Institute of Biosciences, São Paulo State University (UNESP), Rio Claro 13506-900, SP, Brazil; lara.rocha@unesp.br (L.C.R.); gabriela.k.barbosa@unesp.br (G.K.B.); jurandyr.pimentel@unesp.br (J.P.N.); carolina.jacob@unesp.br (C.d.S.J.); 2Department of Sports Traumatology M51, Bispebjerg and Frederiksberg Hospital, IOC Copenhagen Research Center, 1050 Copenhagen, Denmark; andreasknds@gmail.com; 3Department of Anatomy, Institute of Biomedical Science III, University of São Paulo-USP, São Paulo 05508-000, SP, Brazil; watanabe@icb.usp.br

**Keywords:** nuclear domain, sarcomere, telocyte, muscle-tendon perimeter, aquatic training, joint immobilization

## Abstract

The myotendinous junction (MTJ) is the muscle-tendon interface and constitutes an integrated mechanical unit to force transmission. Joint immobilization promotes muscle atrophy via disuse, while physical exercise can be used as an adaptative stimulus. In this study, we aimed to investigate the components of the MTJ and their adaptations and the associated elements triggered with aquatic training after joint immobilization. Forty-four male Wistar rats were divided into sedentary (SD), aquatic training (AT), immobilization (IM), and immobilization/aquatic training (IMAT) groups. The samples were processed to measure fiber area, nuclear fractal dimension, MTJ nuclear density, identification of telocytes, sarcomeres, and MTJ perimeter length. In the AT group, the maintenance of ultrastructure and elements in the MTJ region were observed; the IM group presented muscle atrophy effects with reduced MTJ perimeter; the IMAT group demonstrated that aquatic training after joint immobilization promotes benefits in the muscle fiber area and fractal dimension, in the MTJ region shows longer sarcomeres and MTJ perimeter. We identified the presence of telocytes in the MTJ region in all experimental groups. We concluded that aquatic training is an effective rehabilitation method after joint immobilization due to reduced muscle atrophy and regeneration effects on MTJ in rats.

## 1. Introduction

The myotendinous junction (MTJ) anchors the terminal sarcomeres to the extracellular matrix (ECM). The MTJ is the site of contractile force transmission from the muscle to the tendon, and the higher contact area between the muscle and tendon provides resistance to the muscle contractile force [1].

Joint immobilization promotes inactivity, which leads to muscle atrophy via disuse. Moreover, joint immobilization promotes diverse deleterious effects in the post-immobilization period in the form of impaired muscle performance [2] and reduced myotendinous interface [3,4].

Muscle fibers in normal conditions contain multiple nuclei that are peripherally organized. However, in communicating junctions (such as MTJ), the nuclear position is specialized and demonstrates adaptations associated with mechanical stimulus [5].

Recently, a study identified telocytes as a new element that are adjacent to MTJ [6] in the interstitium. They are characterized by telopods (long and thin projections) and pods (dilated segments) with mitochondria and endoplasmic reticulum [7]. The telopods establish communication and paracrine activity that help different cell types interact, including blood vessels, nerve cells, immune cells, fibrocytes, stem cells, granular cells, and others, under normal and pathological conditions [7,8,9].

The period after joint immobilization is a critical period toward recovery as well as for the mitigation of deleterious effects in muscle fibers and postsynaptic components [10]. Alterations in the ligand proteins of ECM, laminin, and dystrophin in the post-immobilization period [11] reduce the extensibility and plasticity properties of muscle fibers, rendering them susceptible to muscle injuries. However, stimulus in the form of physical activity revealed positive responses to these characteristics [12,13].

In the present study, we investigated the components of the MTJ and their adaptations as well as the associated elements that are triggered following aquatic training (AT) after joint immobilization.

## 2. Results

We observed the adaptations associated with AT and joint immobilization (IM), and their association (IMAT), in the muscle (fiber area and fractal dimension (FD)) and the MTJ region (morphology, nuclear density, sarcomeres, and sarcoplasmatic projections) (Appendix A).

### 2.1. Fiber Area

The AT group had a lower fiber area than the SD group (*p* = 0.0001). The IM group had a lower fiber area than the SD (*p* < 0.0001) and AT groups (*p* < 0.0001). By contrast, the IMAT group had a lower fiber area than the SD group (*p* = 0.0001) and a higher fiber area than the IM group (*p* < 0.0001) (Figure 1).

### 2.2. Fractal Dimension (FD)

The AT (*p* = 0.0015) and IM (*p* = 0.0098) groups had a higher FD than the SD group (Figure 1).

### 2.3. MTJ Nuclear Density

The IM group had a lower MTJ nuclear density than the AT (*p* = 0.0002) and IMAT (*p* = 0.0005) groups. The other groups showed no significant difference (Figure 2).

### 2.4. MTJ Morphology

Through the ultrastructural descriptions of the MTJ, its morphology and plasticity as well as associated elements were revealed, i.e., telocytes (Figure 3). We observed branched sarcoplasmatic invaginations and a telocyte surrounding the MTJ near a blood capillary in the SD group.

In the IM group, we observed a few sarcoplasmatic projections and the presence of telopods near the MTJ as well as a centralized myonucleus in the muscle fiber close to the MTJ (Figure 4). In the AT group, we observed branched sarcoplasmatic invaginations, the presence of a telocyte in ECM and their telopods, and a long sarcoplasmatic invagination connected to a vesicle cluster inside the muscle fiber. Compared with the AT group, the IMAT group had longer sarcoplasmatic projections in addition to telocyte and their associated telopods and a vesicle cluster inside the muscle fiber.

### 2.5. Sarcomeres

The distal sarcomere had longer lengths in the IMAT group than in the IM (*p* < 0.0001) and AT groups (*p* < 0.0001). The AT group had longer proximal than the SD group (*p* < 0.0001). The belly sarcomeres were longer in the IM group than in the SD (*p* = 0.0001) and IMAT groups (*p* = 0.0485); the AT group had longer sarcomeres than the SD group (*p* = 0.0011) (Figure 4).

### 2.6. MTJ Morphometry

The IM group demonstrated a shorter MTJ perimeter than the SD (*p* = 0.001) and AT (*p* = 0.0004) groups. By contrast, the lengths of sarcoplasmatic invagination and evagination in the IM group were not significantly different from the SD group (*p* > 0.05) but were lower than the AT (*p* < 0.0001) and IMAT (*p* < 0.0001) groups.

The AT group showed a higher MTJ perimeter than the IM group (*p* = 0.0004). Moreover, the AT group had longer sarcoplasmatic invagination and evagination lengths than the SD (*p* < 0.0001) and AT groups (*p* < 0.0001).

The IMAT group had longer MTJ perimeter (*p* < 0.0001) and longer sarcoplasmatic (*p* < 0.0001) invaginations than the IM group. In addition, the IMAT group had longer sarcoplasmatic evaginations than the IM (*p* < 0.0001) and AT groups (*p* = 0.001) (Figure 4).

## 3. Discussion

The results of this study revealed the joint immobilization effect of muscle atrophy due to disuse. The study also showed the effect and repercussion of AT as a rehabilitation method for the muscle belly and the muscle-tendon complex as well as other parameters, including the nucleus, sarcomere, contact perimeter, and telocyte.

The lower fiber observed after joint immobilization indicates muscle atrophy due to disuse [14] and a higher FD value, indicating a nucleus with higher organization complexity similar to muscle injury results [15].

FD analysis is a model of organization complexity for evaluating diverse tissue components [16,17]. Through AT, we observed a higher nucleus FD, indicating a higher cellular activity because of the enhancement of mechanical stress due to physical exercise; this adaptive response is favorable for better muscular performance [18].

AT after joint immobilization did not lead to a higher organization complexity of the nucleus, although a higher fiber area was observed than in the IM group. These results indicate that training contributes to muscle readaptation after joint immobilization and associated structures, however without full return to its pre-immobilization morphology; these results were also observed by Nascimento et al. [19].

Regarding sarcomeres, we observed different lengths, primarily in the MTJ region. Moo et al. described sarcomere length adaptation in different muscle sites and confirmed nonuniformity in the same muscle fibers [18].

In the joint immobilization, we observed longer sarcomeres in the muscle belly. This finding is directly associated with the limb position during the immobilization period. In the short position the length may increase, although the amount of sarcomeres over the fiber is lower. By contrast, the immobilized limb in the extended position may have an inverse effect [20,21,22].

AT after joint immobilization demonstrated a higher sarcomeric variation in the MTJ; we associated this finding with increased tension in the region due to physical exercise, especially after a weakness moment from muscle disuse [23,24]. In the IMAT group, this led to longer lengths in the distal and belly sarcomeres. This was possibly because of sarcomeric disposition prior to muscle activity (AT) and inactivity (joint immobilization) protocols [25]. Beyond physical activity, stretching presents beneficial effects for the sarcomeres, including the improved reduction of actin and myosin filaments as well as reduction in muscle deterioration [26].

The myotendinous region has a different myonuclear domain due to the specialty of the muscle-tendon transmission of shear force [27]. Based on the results, we must consider the possible presence of a nucleus associated with muscle tissue in the form of a myonucleus, satellite cells, and stromal cells and nuclei associated with tendinous tissue in the form of telocytes, fibroblasts, and tenocytes.

After joint immobilization we demonstrated a lower nuclear density in the MTJ region. The inactivity can cause muscle atrophy owing to myonuclear apoptosis [28,29]. The nuclear domain of the MTJ is essential to the myotendinous interface function, and the misplaced nuclei can result in muscle dysfunction [30].

Joint immobilization revealed a smaller MTJ perimeter, representing a lower surface of interaction between the muscle and tendon tissues. These results suggest a reduction in the transmission of the tension that is applied in the fiber area to the interface, leading to predisposition for shear-force injuries [31,32].

The general morphology of the MTJ is modified with atrophy induced by joint immobilization; we observed a centralized myonucleus in the muscle fiber in this group. This effect is commonly observed in myofibrillar development, inflammation process recovery, and myopathies [16,30]. However, myonuclei localization is determined in front of the region function [5], and we observed this effect due to remodeling stimuli associated with joint immobilization.

AT after joint immobilization demonstrated that nuclear density in the MTJ region of the IMAT group is similar to that in the MTJ region of the AT. Machida and Booth [33] found that the recovery period after unloading leads to an increase in the myonuclei amount, thereby stimulating satellite cells to improve their activation and differentiation. They also discussed the influence of the higher Ca²^+^ influx for this recovery, which in our study possibly occurred before AT.

The structural components of the MTJ presented with varying adaptations with respect to the protocols; the results demonstrated alterations in the contact surface of the muscle and tendon. With AT we observed a greater MTJ perimeter, similar to that observed by Sierra et al. [24]. Moreover, Jacob et al. [34] observed a similar result with training and the associated rehabilitation process in the MTJ region and muscle quality in rats, which reduces the predisposition to injury in the MTJ region.

The MTJ region demonstrated broad reorganization with AT after joint immobilization; it conferred a longer MTJ perimeter and longer sarcoplasmatic invaginations and evaginations, resulting in a stable force transmission as well as reduced stress absorption and injury [35].

We observed telocytes and their telopods (longer and thin projections) surrounding the MTJ, which were recently identified in the region [6]. The MTJ shows a higher adaptive response to differing stimuli, e.g., aging [15,21], physical exercise [4,36], and obesity [37]. Morphometric adaptation and the presence of telocytes reaffirm the hypothesis that performance in an effective manner in a distinct stimulus is needed for MTJ adaptation beyond the function of tissue regeneration support [38].

In this study, the groups associated with AT (AT and IMAT group) had telopods adjacent to sarcoplasmatic invaginations and vesicles, correlating with the paracrine/juxtacrine activity of telocytes in the MTJ. The telocytes associated with the paracrine/juxtacrine activity have already been found with the atrophic factor [39] and shown to be associated with satellite cells for regeneration support [40].

## 4. Materials and Methods

### 4.1. Animals

We classified 44 male Wistar rats aged 90 days into 4 groups (n = 11): sedentary group (SD), underwent no protocol; IM group, underwent a joint immobilization protocol for 10 days; AT group, underwent an aquatic training protocol (4 weeks); and immobilization/aquatic training (IMAT) group, underwent the joint immobilization protocol for 10 days and later the AT protocol. The rats were placed in cages (33 × 40 × 16 cm; n = 4), under a controlled temperature of 23 °C ± 2 °C and a light/dark cycle of 12 h with access to food and water ad libitum. The Committee on Ethics in Animal Use (CEUA) of the Biosciences Institute of the São Paulo State University (UNESP) (no. 2018/1220, 06/06/2018) approved this study.

### 4.2. IM Protocol

The IM and IMAT groups were anesthetized (ketamine 95 mg/kg and xylazine 12 mg/kg, intraperitoneal (i.p. injection)), and an immobilizing device made of steel mesh, cotton, and tape was applied to the tibiotarsal joint of the right posterior limb in a short position for 10 days [11]. After the removal of joint immobilization, we collected samples from the IM group. The IMAT group underwent the AT protocol.

### 4.3. AT Protocol

The AT and IMAT groups underwent the AT protocol in a rectangular tank, which were separated individually by tubes, in a depth of 40 cm water at 31 °C, and the animals swam freely in the tube. The protocol comprised 5 weekly sessions of 60 min each; a total of 20 sessions were completed in 4 weeks. A load of 3% of the rats’ body mass was fixed to the thorax, which was corrected weekly [34,41].

### 4.4. Light Microscopy

The rats of each group (n = 5) were euthanized with an anesthetic overdose (ketamine 200 mg/kg and xylazine 50 mg/kg, i.p. injection). Gastrocnemius muscle samples were cryo-fixed and stored at −80 °C. We collected transversal sections of the belly muscles and longitudinal sections of the MTJ region (10-µm-thick Cryostat HM 505E, MICROM^TM^). These were then stained using hematoxylin-eosin (HE) [42,43]. Light microscopy images were acquired using a Leica DM750^TM^ (Heerbrugg, Switzerland) at 200× magnification (Figure 5). The muscle fiber areas (n = 60) were measured from the HE-stained images using ImageJ software (National Institutes of Health, Bethesda, MD, USA).

FD analysis was performed to identify the nuclei organizational complexity of the belly of gastrocnemius muscle (n = 14 images). FD is a geometric analysis of organizational complexity using pixel distribution in the image space and a compelling analysis of tissue organizational patterns. The images were binarized using ImageJ software (National Institutes of Health, Bethesda, MD, USA); the FD value was established on a predetermined 0–2 scale, in which 2 denoted higher organizational complexity [16,44].

### 4.5. Transmission Electron Microscopy

The rats of each group (n = 3) were euthanized with an anesthetic overdose (ketamine 200 mg/kg and xylazine 50 mg/kg, i.p. injection). The MTJ sample of the gastrocnemius muscle (3 mm³) was dissected and immersed in a *modified Karnovsky* solution for 48 h at 4 °C. Then, the samples were post-fixed in 1% osmium tetroxide solution for 2 h at 4 °C, dehydrated via an increasing concentration series of alcohol solutions, and embedded in resin (Low Viscosity Embedding Media Spurr’s Kit from Electron Microscopy Sciences). The ultrathin (60 nm) ultramicrotome sections were subsequently collected on 200-mesh copper grids (Sigma-Aldrich™) and stained with 4% uranyl acetate solution and 0.4% aqueous lead citrate [45,46]. The grids were examined and MTJ micrographs were obtained using a JEOL 1010 transmission electron microscope (Peabody, MA, USA) (Figure 5).

Through the transmission micrographs of the MTJ region, we measured the sarcoplasmatic invagination (n = 80) and invagination (n = 80) lengths [33], the MTJ perimeter (n = 13) delimited by the basal lamina length, and a straight base of 2 µm length. We also measured the lengths of the sarcomeres present in the MTJ region (n = 50), both distal (last sarcomere of myofilament) and proximal (penultimate sarcomere), as well as belly muscle sarcomeres [20,24], via ImageJ software (National Institutes of Health, Bethesda, MD, USA).

### 4.6. Immunohistochemistry

The rats of each group (n = 5) were euthanized with an anesthetic overdose (ketamine 200 mg/kg and xylazine 50 mg/kg, i.p. injection). Gastrocnemius muscle samples were cryo-fixed and stored at −80 °C. Then, longitudinal sections of the MTJ region were acquired (Cryostat HM505 E, MICROM^TM^). The samples were washed thrice with phosphate-buffered saline (PBS) containing 1% bovine serum albumin (BSA) and permeabilized with Triton X-100 0.1% for 20 min. The treated samples were then washed again with PBS three times.

For MTJ identification and nuclear analysis (sections of 100-µm thickness), immunostaining was performed using Alexa Fluor™ 488 Phalloidin diluted in PBS (1:600, Invitrogen, A12379) for 30 min for the identification of actin filaments (F-actin), and nucleus staining was performed with 4′,6-diamidino-2-phenylindole (Molecular Probes, Eugene, P36935). Images for histological analysis were obtained using a confocal laser scanning microscope (Leica^TM^ TCSSP5) and analyzed with a Olympus BX61 Fully Motorized Fluorescence Microscope (Shinjuku, Japan) (Figure 5). We measured the nuclear density in the MTJ region (n = 14 images) in an area of 91.3 mm² (1000× magnification).

### 4.7. Immunofluorescence

For telocyte identification, 10-µm-thick sections were collected, immunostained with primary antibody (CD34, 1:1000, IgG polyclonal, Invitrogen, PA5-85917), and diluted in PBS with 1% BSA. After two washes in PBS, the stained slides were incubated with goat anti-rabbit secondary antibody conjugated with Alexa Fluor 594 (1:1000, IgG, Invitrogen, A-11012) and diluted in PBS with 1% BSA for 60 min. Nuclei were stained with 4′,6-diamidino-2-phenylindole (DAPI, Molecular Probes, Eugene, P36935). The histological sections were analyzed with a Olympus BX61 Fully Motorized Fluorescence Microscope (Shinjuku, Japan) at a magnification of 200×. We obtained a differential interference contrast image to visualize the muscle-tendon interface.

### 4.8. Statistical Analysis

We evaluated the normality of the obtained data with a Shapiro-Wilk test. We then performed the Kruskal-Wallis test with Dunn’s post-hoc test (significance, *p* < 0.05) for all analyses.

## 5. Conclusions

We conclude that joint immobilization resulted in muscle atrophy, thereby lowering the myotendinous interface and the adjacent nuclear density. The AT after joint immobilization reestablished the muscle fibers area, increased the MTJ perimeter, the distal sarcomeres, and increased the MTJ nuclear density. Moreover, the results suggest that MTJ plasticity along with the telocyte is associated with diverse elements adjacent to muscle and tendon tissues in the muscle atrophy model and AT. This investigation presents the results of muscle atrophy rehabilitation via AT in the MTJ in rats, but more studies must be done to understand the effects of these protocols in humans and their application.

## Figures and Tables

**Figure 1 ijms-22-06983-f001:**
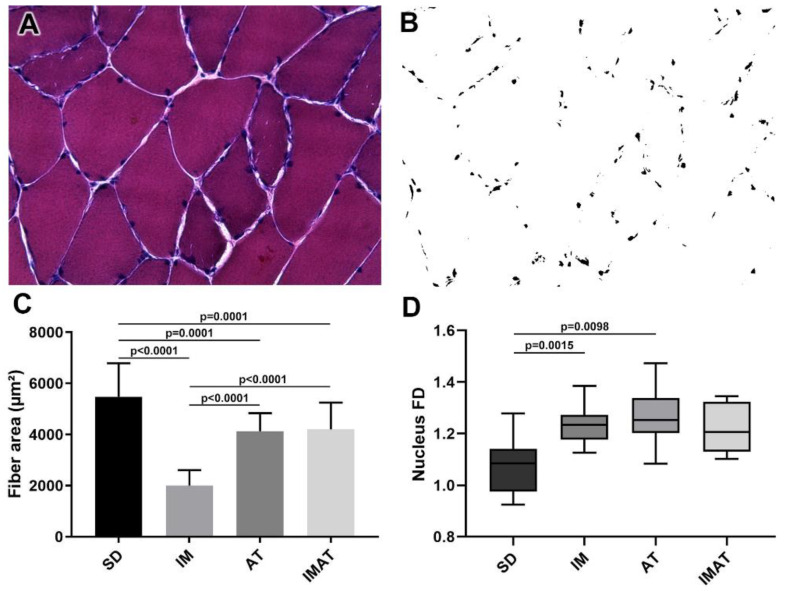
Analysis of muscle fibers of the belly of gastrocnemius muscle of *Wistar* rats. (**A**) Light microscopy of belly muscle in hematoxylin-eosin staining (HE), magnification: 400×. (**B**) Binarized image of HE staining with nucleus highlight for fractal dimension (FD) analysis. (**C**) Mean ± standard deviation of muscle fibers area (µm²) of sedentary (SD), immobilization (IM), aquatic training (AT), and immobilization/aquatic training (IMAT); SD ≠ AT (*p* = 0.0001), SD ≠ IM (*p* < 0.0001), SD ≠ IMAT (*p* = 0.0001), AT ≠ IM (*p* < 0.0001), and IM ≠ IMAT (*p* < 0.0001). (**D**) Box plot of nucleus FD analysis of groups; SD ≠ AT (*p* = 0.0015) and SD ≠ IM (*p* = 0.0098).

**Figure 2 ijms-22-06983-f002:**
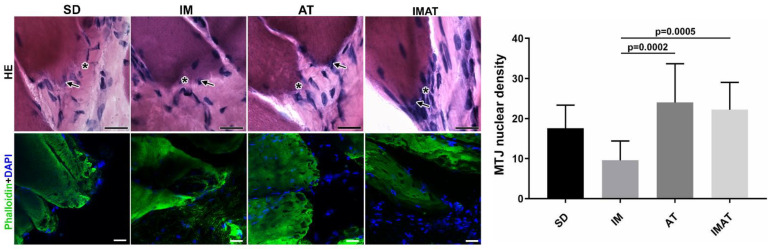
Light microscopy of hematoxylin-eosin staining (HE) of MTJ (*) and adjacent nucleus (arrow) of sedentary (SD), immobilization (IM), aquatic training (AT), and immobilization/aquatic training (IMAT) from gastrocnemius muscle of *Wistar* rats. Bars: 20 µm. Immunohistochemistry of MTJ region of F-actin (Phalloidin) and nucleus (DAPI). Bars: 20 µm. Mean ± standard deviation of MTJ nuclear density (unity/91.3 mm²) of groups; AT ≠ IM (*p* = 0.0002) and IM ≠ IMAT (*p* = 0.0005).

**Figure 3 ijms-22-06983-f003:**
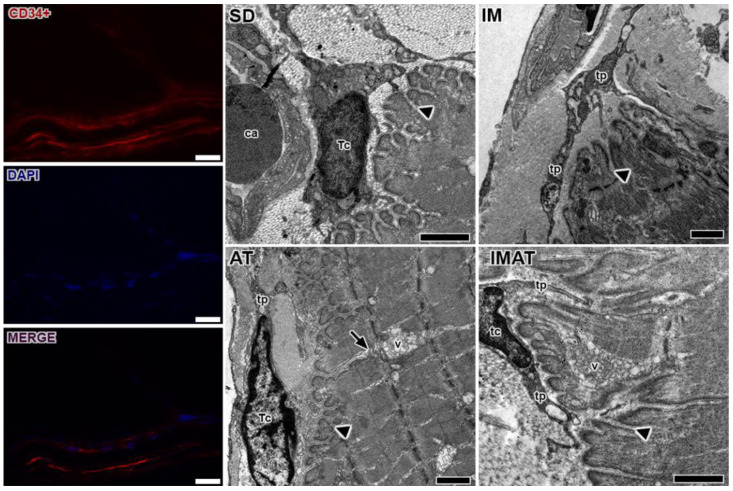
Telocytes analysis of MTJ region with immunostaining (CD34+), nucleus (DAPI), and the image with associated elements (MERGE). Bars: 20 µm. Transmission electron microscopy of MTJ of sedentary (SD), immobilization (IM), aquatic training (AT), and immobilization/aquatic training (IMAT) from gastrocnemius muscle of *Wistar* rats. In the myotendinous region the presence of blood capillary (ca) and telocyte (tc) with their projections the telopods (tp) is observed; these elements were observed adjacent to sarcoplasmatic invaginations (arrowhead), and in the AT group a prolonger branch of sarcoplasmatic invagination (arrow) with association with a vesicle group (v) is observed. Bars: 1 µm.

**Figure 4 ijms-22-06983-f004:**
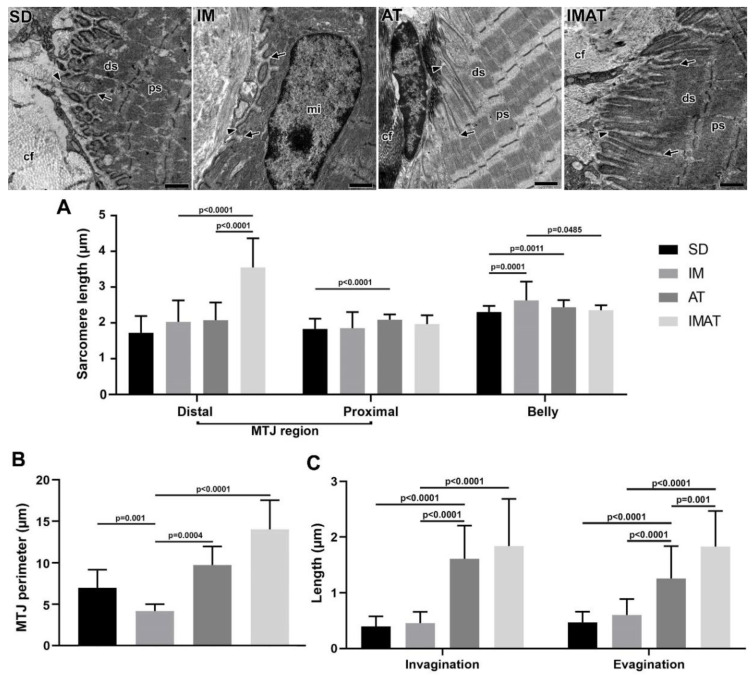
Transmission electron microscopy of MTJ region of sedentary (SD), immobilization (IM), aquatic training (AT), and immobilization/aquatic training (IMAT) from gastrocnemius muscle of *Wistar* rats. In the ECM collagen fibers (cf) organized in transversal and longitudinal form were observed, as well as the sarcoplasmatic invaginations (arrowhead) and evaginations (arrow) associated to distal (ds) and proximal (ps) sarcomeres; in IM group a centralized myonucleus (mi) in the muscle fiber was observed. Bars: 1 µm. (**A**) Mean ± standard deviation of sarcomere length (µm) localized in the MTJ region, distal and proximal, and of the belly muscle of groups; distal sarcomere: IM and AT ≠ IMAT (*p* < 0.0001); proximal sarcomere: SD ≠ AT (*p* < 0.0001); belly sarcomere: IM ≠ SD (*p* = 0.0001), AT ≠ SD (*p* = 0.0011), and IM ≠ IMAT (*p* = 0.0485). (**B**) Mean ± standard deviation of MTJ perimeter determined in the contact area of 2 µm between sarcoplasmatic invagination and evagination of groups; SD ≠ IM (*p* = 0.001), AT ≠ IM (*p* = 0.0004), and IM ≠ IMAT (*p* < 0.0001). (**C**) Mean ± standard deviation of length of sarcoplasmatic invagination and evaginations of groups; invaginations: AT ≠ SD (*p* < 0.0001), AT ≠ IM (*p* < 0.001), and IMAT ≠ IM (*p* < 0.0001); evaginations: AT ≠ SD (*p* < 0.0001), AT ≠ IM (*p* < 0.0001), IMAT ≠ IM (*p* < 0.0001), and IMAT ≠ AT (*p* = 0.001).

**Figure 5 ijms-22-06983-f005:**
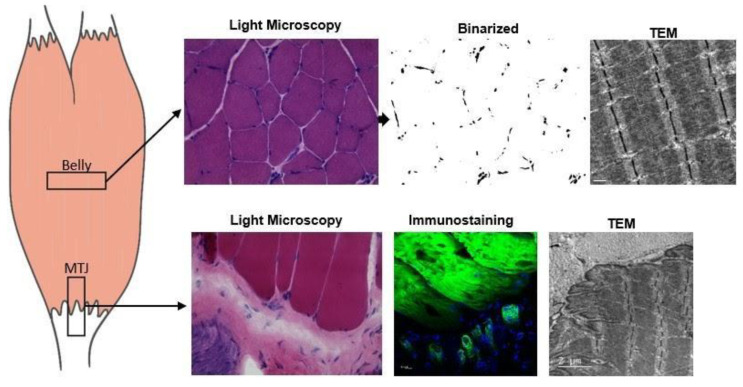
Illustrative image of the gastrocnemius muscle of *Wistar* rats, with the demarcation of regions (squares) from where the sections and samples of the muscle belly and myotendinous junction (MTJ) regions were performed. The samples of muscle belly collected in transverse sections were processed for the light microscopy technique and stained with hematoxylin-eosin (HE) and then binarized using the ImageJ software for cell nuclei identification (magnification: 400×); the samples of muscle belly collected in longitudinal sections were analyzed using the transmission electron microscopy (TEM) technique to identify sarcomeres in series (magnification: 15,000×). The samples of the myotendinous region collected in longitudinal sections were processed for light microscopy technique and stained with HE (magnification: 400×), and immunostaining with Alexa Fluor™ 488 Phalloidin and DAPI (magnification: 1000×); the TEM technique was used to identify the MTJ ultrastructures (magnification: 10,000×).

## Data Availability

The data presented in this study are available in the article.

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
