# Peer review of "Aquatic Training after Joint Immobilization in Rats Promotes Adaptations in Myotendinous Junctions"

_ijms, 2021, doi:10.3390/ijms22136983_

Round 1

Reviewer 1 Report

Well done for incorporating the necessary changes into the manuscript.

Author Response

We thank you for all your compliments and considerations. The English language was revised and improved by Crimson Interactive (Enago- INQ-4152372721).

Reviewer 2 Report

I found the article very interesting. It has good study design and interesting findings. However, I would suggest rearrangement of the discussion. It should be more structural and aimed on logical process leading to conclusions. 

For example, please discuss muscle re-adaptation issue in lines 157-159 in relation to reference cited[19].

Please, try not to enumerate all your results in the discussion section, discuss your results in relation to the literature instead. 

Author Response

We thank you for all your compliments and considerations. The discussion topic was reorganized according to the proposal. We understand that some statements may seem not directly related to the result, because we care not to be left in a speculative gap. The English language was revised and improved by Crimson Interactive (Enago- INQ-4152372721).

Reviewer 3 Report

Aquatic training after joint immobilization in rats promotes adaptations in myotendinous junctions

Journal: IJMS (ISSN 1422-0067)

Manuscript ID: ijms-1277361

The main aim of this study was to investigate the components of MTJ and their adaptations as well as the associated elements that are triggered following aquatic training (AT) after joint immobilization.

The approach of the study appears very original and the contents of the manuscript are quite interesting by his methodology and through the tools of quantification used. By the interesting methodology presented, the authors showed an interesting rigor of analysis.

The manuscript reads smoothly and is easy to understand.  The aims, scope, and results of the study are clearly stated.  I have very much enjoyed reading this paper. I find it interesting and clearly written, and satisfying also all the other publication criteria of the “International Journal of Molecular Sciences”. The study provides a very valuable addition to this line of research, and adds relevantly to the subject with additional original findings. I thus find that this paper definitively delivers results that will surely be of interest to the readership of the “International Journal of Molecular Sciences”.  I recommend the publication of this interesting paper after the following remark:

The conclusion is not sufficiently developed and requires further clarification. How the results obtained can be used? The authors' notes on this point are not sufficient and require more development. This,  even though a weak note in the abstract is noted by the authors   “We concluded that aquatic training is an effective rehabilitation method after joint immobilization due to reduced muscle atrophy and regeneration effects on MTJ”

Author Response

We thank you for all your compliments and considerations. In the conclusion topic, we add the main factors to be considered in our experimental model presented and thus highlight its effects in humans. The English language was revised and improved by Crimson Interactive (Enago- INQ-4152372721).

This manuscript is a resubmission of an earlier submission. The following is a list of the peer review reports and author responses from that submission.

Round 1

Reviewer 1 Report

The MS “Aquatic rehabilitation after joint immobilization in rats promotes adaptations in myotendinous junctions” by Rocha and colleagues is aimed on the investigation how physical exercises help the organism to recover after immobilization. Despite obvious importance of the question raised, the overall conclusions seem to me quite predictable. The work described is more to deal with the narrow aspects of adaptive physiology and it is worse publishing in a more specialized journal.

I would suggest to put exact values of all the measured parameters (mean and SD) in the text of the MS, or include the corresponding tables. The p-values of statistical significance given with high precision are not supported by the diagrams only.

English language can be improved.

Reviewer 2 Report

The manuscript handles a relevant topic for rehabilitation sciences.

In the introduction section, please clearly state the study was carried out on rats and explain that the study outcomes may have relevance to humans.

I would also recommend that the authors adjust their statements and referencing based on the various species throughout the manuscript

Readers might get confused that the experiments were carried out on rats, but some citations refer to human studies. Overall recommendations for the figures: please add to the figure legends the muscle name and the species name (rat).

Reviewer 3 Report

In this manuscript, the authors showed that aquatic training affected the morphology o myotendinous junction in rats. They examined myotendinous junction by microscope and ultramicroscope and found increased MTJ nuclear density, sarcomere length at the distal region, MTJ perimeter, invagination length. However, the data are not clear because the authors did not show all images they analyzed. In addition, in each analysis, it is very unclear which region was analyzed and how it was measured. The reviewer also felt that the introduction and discussion sentences were very fragmented, and the results sentences were too descriptive to give the reader the full meaning of the observations.

The authors concluded that "aquatic training is an effective rehabilitation method after joint immobilization due to reduced muscle atrophy and regeneration effects on MTJ". However, the reviewer felt that the effect of aquatic training is minimal, because of large error bars in each experimental result. It would be better to show the values of each sample with a dot, rather than just a graph, to clearly show the variability of the experimental results. It is also recommended that the authors should ask a native speaker to check for the manuscript.